# The effect of diabetes mellitus on COVID-19 mortality among patients in a tertiary-level hospital in Bandung, Indonesia

Maya Kusumawati[1], Raspati Cundarani Koesoemadinata[2,3]*, Zuhaira Husna Fatma[2], Evan Susandi[4], Hikmat Permana[1], Nanny Natalia Mulyani Soetedjo[1,2], Arto Yuwono Soeroto[5], Begawan Bestari[6], Basti Andriyoko[2,7], Bachti Alisjahbana[2,8], Yovita Hartantri[2,8]

1 Division of Endocrinology and Metabolism, Department of Internal Medicine, Faculty of Medicine, Dr Hasan Sadikin General Hospital, Universitas Padjadjaran, Bandung, Indonesia, 2 Research Center for Care and Control of Infectious Disease, Universitas Padjadjaran, Bandung, Indonesia, 3 Indonesian Society for Clinical Microbiology, Jakarta, Indonesia, 4 Department of Internal Medicine, Faculty of Medicine, Dr Hasan Sadikin General Hospital, Universitas Padjadjaran, Bandung, Indonesia, 5 Respirology and Critical Illness Division, Department of Internal Medicine, Faculty of Medicine, Dr Hasan Sadikin General Hospital, Universitas Padjadjaran, Bandung, Indonesia, 6 Division of Gastroentero Hepatology, Department of Internal Medicine, Faculty of Medicine, Dr Hasan Sadikin General Hospital, Universitas Padjadjaran, Bandung, Indonesia, 7 Molecular Biology Division, Department of Clinical Pathology, Faculty of Medicine, Dr Hasan Sadikin General Hospital, Universitas Padjadjaran, Bandung, Indonesia, 8 Division of Tropical and Infectious Disease, Department of Internal Medicine, Faculty of Medicine, Dr Hasan Sadikin General Hospital, Universitas Padjadjaran, Bandung, Indonesia

* r.c.koesoemadinata@unpad.ac.id

**Data Availability Statement:** Minimal data set have been shared as a Supporting information File (S1 File).

## Abstract

Immune system dysregulation in people with diabetes mellitus (DM) increases the risk of acquiring severe infection. We compared the clinical characteristics and laboratory parameters of coronavirus disease 2019 (COVID-19) patients with and without DM and estimated the effect of DM on mortality among COVID-19 patients. A retrospective cohort study collecting patients' demographic, clinical characteristics, laboratory parameters and treatment outcomes from medical records was conducted in a hospital in Bandung City from March to December 2020. Univariable and multivariable logistic regression was performed to determine the association between DM and death. A total of 664 COVID-19 patients with positive real-time reverse transcription polymerase chain reaction for severe acute respiratory syndrome coronavirus 2 were included in this study, of whom 147 were with DM. Half of DM patients presented HbA1c $\geq$10%. DM patients were more likely to present with comorbidities and severe to critical conditions at admission (P <0.001). Laboratory parameters such as neutrophil-lymphocyte count ratio, C-reactive protein, D-dimer, ferritin, and lactate dehydrogenase were higher in the DM group. In the univariate analysis, variables associated with death were COVID-19 severity at baseline, neurologic disease, DM, age $\geq$60 years, hypertension, cardiovascular disease, and chronic kidney disease. DM remained associated with death (aOR 1.82; 95% CI 1.13–2.93) after adjustment with sex, age, hypertension, cardiovascular disease, and chronic kidney disease. In conclusion, COVID-19 patients with DM are more likely to present with a very high HbA1c, comorbidities, and severe-critical

**Funding:** The authors received no specific funding for this work.

**Competing interests:** The authors have declared that no competing interests exist.

illness. Chronic inflammation in DM patients may be aggravated by the disruption of immune response caused by COVID-19, leading to worse laboratory results and poor outcomes.

## Introduction

The World Health Organization (WHO) declared a pandemic of coronavirus disease 2019 (COVID-19) in March 2020 [1]. There were already 278 million confirmed cases with 5.4 million deaths globally until the end of December 2021 [2]. While during the same period, Indonesia reported over 4 million cases and nearly 150 thousand deaths [3].

There has been evidence of more severe COVID-19 in people with diabetes mellitus (DM) [4]. Immune system dysregulation and chronic hyperglycaemia are suspected of causing poorer outcomes in COVID-19 patients with diabetes mellitus [5–7]. However, clinical and laboratory features associated with poor outcomes in COVID-19 patients may vary in different settings. Therefore, we aim to gain more understanding of the clinical characteristic and laboratory parameter differences between diabetic COVID-19 patients and nondiabetic COVID-19 patients and to measure the strength of the association between diabetes with mortality in COVID-19 patients in our hospital in Bandung City, Indonesia.

## Methods

### Study design and setting

This retrospective cohort study included COVID-19 patients admitted to Dr Hasan Sadikin General Hospital, Bandung City, Indonesia, between March and December 2020. We collected data from medical records of adult patients with positive real-time reverse transcription polymerase chain reaction (rRT-PCR) for severe acute respiratory syndrome coronavirus 2 (SARS-Cov-2) in inpatient care during the study period. Pregnant women, people living with HIV/AIDS, and patients with autoimmune diseases were excluded from the study. This study was reviewed and approved by Universitas Padjadjaran Ethical Committee (ethical clearance No. LB.02.01/X.6.5/226/2020). After consideration of the research method used (retrospective study), the need for informed consent from the study subjects was waived. All aspects of this study were conducted adhering to the principles of the Helsinki Declaration.

### Data collection

Data collected included patients' demographic (age and sex), clinical symptoms (fever, cough, sore throat, headache, dyspnoea, coryza, fatigue, anosmia, diarrhoea and nausea/vomiting), duration from onset of illness to hospital admission, and comorbidities (no comorbidities, hypertension, cardiovascular disease, chronic kidney disease, liver disease, malignancies, chronic pulmonary comorbidities, neurological disease, and other). Body mass index (BMI), oxygen saturation (%), radiological results (chest x-ray), and laboratory results (glycated haemoglobin (HbA1c), neutrophil-lymphocyte count ratio (NLCR), random blood glucose (RBG), C-reactive protein (CRP), ferritin, lactate dehydrogenase (LDH), and D-dimer) were also collected from the medical records. Other clinical characteristics measured included disease severity during admission and treatment outcomes. Data from medical records were entered into an electronic database (REDCap) [8].

BMI was categorized using cut-points for the Asian populations: underweight ($<18.5$ kg/m$^2$), normal (18.5–22.9 kg/m$^2$), overweight (23–24.9 kg/m$^2$), and obese ($\geq25.0$ kg/m$^2$) [9]. Disease severity was classified according to the WHO treatment guidelines: mild (symptomatic

patient without pneumonia or hypoxia), moderate (clinical symptoms of pneumonia without signs of severe pneumonia), severe (pneumonia symptoms presenting with respiratory rate >30 times per minute, severe respiratory distress, or oxygen saturation <90%), and critical illness (patients with acute respiratory distress, sepsis, or septic shock) [10]. DM status of the patients was determined following the Indonesian Endocrinologist Association's guidelines in which the patient must present with a fasting blood glucose level ≥126 mg/dl or random blood glucose level ≥200 mg/dl with classical hyperglycaemia symptoms (polydipsia, polyuria, weight loss), or HbA1c ≥6.5%, or previous history of DM [11].

## Statistical analysis

Baseline characteristics of COVID-19 patients with DM were compared with COVID-19 patients without DM. Chi-square and Fisher's exact tests were carried out for categorical data. For the continuous and numerical variables, we performed a normality test using the Shapiro-Wilk test, then proceeded using an independent t-test for normally distributed data or the Mann-Whitney test for skewed distributed data. A P-value of <0.05 represents statistical significance. We described the HbA1c values of DM patients during admission by a bar graph. The association between DM and mortality was examined using univariable and multivariable logistic regression. Multivariable analysis included age, sex, and comorbidities associated with chronic DM (hypertension, cardiovascular disease, and chronic kidney disease). These are variables that existed before the patient was infected by COVID-19. Age was categorized as below 60 or 60 years old and over. The odds ratio (OR) and corresponding 95% confidence interval (CI) were presented as the effect size. Statistical analyses were conducted using Stata version 13 (StataCorp, College Station, TX, USA).

## Results

A total of 692 COVID-19 patients with positive rRT-PCR were admitted to the Hasan Sadikin General Hospital during the study period. Twenty-eight patients were excluded due to the following reasons: twenty-three were pregnant, three patients had an autoimmune disease, and two patients were HIV-positive, leaving 664 (93.4%) patients included in the analysis. Of the included patients, 147 (21.1%) were patients with diabetes, and 517 (77.9%) were nondiabetic patients (Fig 1).

There were slightly more males than females, with a similar proportion in both DM and non-DM groups. Patients in the DM group were older than the non-DM group, with a median age of 60 years (IQR 49–67) and 49 years (IQR 36–59), respectively. About half (54.4%) of the study subjects had underlying comorbidities. Of those with comorbidities, DM patients were more likely to present with hypertension (P <0.001), cardiovascular disease (P <0.04), and chronic kidney disease (P <0.001). The presence of liver disease, malignancy, chronic pulmonary disease, neurological disease, and other comorbidities did not significantly differ between the two groups (P >0.05) (Table 1).

The overall median onset of symptoms was six days. There is a difference between the two groups, although not statistically significant (P = 0.06), with the nondiabetic group feeling symptoms earlier than the diabetic group prior to hospital admission. Clinical symptoms were also similar in both groups, except for cough and dyspnoea, which occurred more often in DM patients. There was no significant difference in the median body mass index (BMI) (P = 0.65). Median oxygen saturation was higher in the non-DM group compared to the DM group (P <0.001). Pneumonia was also found to be more frequent in the chest x-ray of DM patients compared to non-DM patients (P <0.001). In addition, DM patients are more likely to present

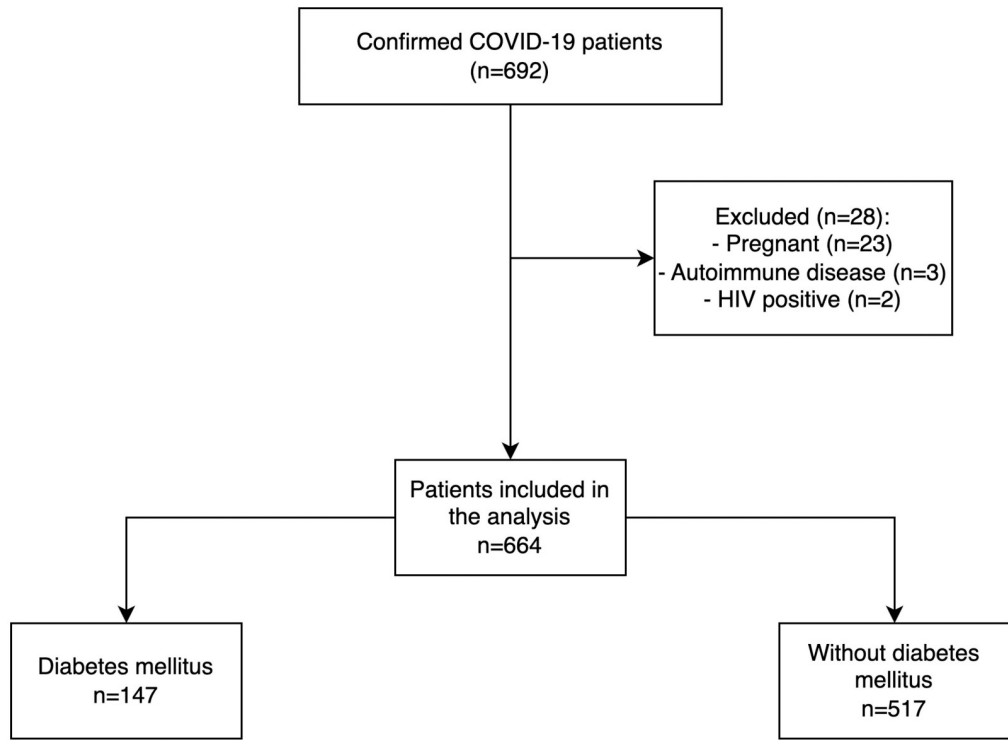

**Fig 1. Patients' flowchart.**

with severe or critical disease (P <0.001) on admission, and more DM patients died (27.2% vs 13.0%, P <0.001) (Table 1).

Of all DM patients included, HbA1c was measured for 118 (80.3%) of them. About half (53.4%) of the DM patients had an HbA1c of ≥10.0%, while there were only 12 (10.2%) patients with HbA1c <7.0% (Fig 2). All laboratory parameters other than HbA1c measured were higher in the DM group than in the non-DM group (Table 2). The median NLCR in the DM group was 5.61 (IQR 3.47–9.75) compared to 3.79 (IQR 2.18–6.50) in the non-DM group (P <0.001). The median CRP were 7.26 mg/dL (IQR 3.45–12.74) and 2.25 mg/dL (IQR 0.46–8.87) in the DM group and non-DM group (P <0.001), respectively. The median D-dimer in the DM group was 1.02 ng/mL (IQR 0.54–2.14) vs 0.69 ng/mL (IQR 0.38–1.55) in the non-DM group. The median LDH were 352 u/L (IQR 256–444) and 283 u/L (IQR 201–407) in DM and non-DM groups, respectively.

Patient characteristics associated with death in the univariable analysis were older age (P <0.001), low level of oxygen saturation (P <0.001), pneumonia on chest x-ray (P <0.001), and severe or critical condition during admission (P <0.001). Mortality is also higher in patients with multiple comorbidities (2 or more comorbidities). Among the comorbidities, DM (P <0.001), hypertension (P = 0.004), cardiovascular disease (P = 0.001), chronic kidney disease (P = 0.003), and neurological disease (P <0.001) were associated with death (Table 3). In addition, there were 24 patients with neurologic disease; stroke (n = 9), encephalopathy or structural stroke (n = 2), malignancy (n = 3), meningitis (n = 2), and unspecified (n = 8). Even though severe or critical conditions during admission and neurologic disease were strongly associated with mortality, these variables were not included in the multivariable analysis because they were considered to be in the causal pathway of mortality and may not be associated with DM.

**Table 1. Sociodemographic and clinical characteristics of patients.**

| Characteristics | Total (n = 664) | Diabetes Mellitus | | |
|---|---|---|---|---|
| | | Yes (n = 147) | No (n = 517) | P |
| Male | 357 (53.8) | 79 (53.7) | 278 (53.8) | 1.00 |
| Median age (IQR), years | 52 (39–62) | 60 (49–67) | 49 (36–59) | <0.001 |
| Comorbidities* | | | | |
| No comorbidities | 303 (45.6) | 0 (0.0) | 303 (58.6) | <0.001 |
| Hypertension | 185 (27.9) | 60 (40.8) | 125 (24.2) | <0.001 |
| Cardiovascular disease | 76 (11.4) | 24 (16.3) | 52 (10.1) | 0.04 |
| Chronic kidney disease | 48 (7.2) | 24 (16.3) | 24 (4.6) | <0.001 |
| Liver disease | 4 (0.6) | 1 (0.7) | 3 (0.6) | 0.89 |
| Malignancies | 16 (2.4) | 1 (0.7) | 15 (2.9) | 0.12 |
| Chronic pulmonary disease | 20 (3.0) | 1 (0.7) | 19 (3.7) | 0.06 |
| Neurological disease | 24 (3.6) | 6 (4.1) | 18 (3.5) | 0.73 |
| Other comorbidities | 25 (3.8) | 2 (1.4) | 23 (4.4) | 0.08 |
| Median symptom onset (IQR), days | 6 (3–8) | 7 (4–10) | 6 (3–8) | 0.06 |
| Symptoms | | | | |
| Fever | 465 (70.0) | 108 (73.5) | 357 (69.0) | 0.30 |
| Cough | 478 (72.0) | 121 (82.3) | 357 (69.0) | 0.002 |
| Sore throat | 124 (18.7) | 20 (13.6) | 104 (20.1) | 0.07 |
| Headache | 85 (12.8) | 23 (15.6) | 62 (12.0) | 0.24 |
| Dyspnoea | 362 (54.5) | 107 (72.8) | 255 (49.3) | <0.001 |
| Coryza | 100 (15.1) | 17 (11.6) | 83 (16.0) | 0.18 |
| Fatigue | 98 (14.8) | 27 (18.4) | 71 (13.7) | 0.16 |
| Anosmia | 27 (4.1) | 8 (5.4) | 19 (3.7) | 0.34 |
| Diarrhoea | 34 (5.1) | 4 (2.7) | 30 (5.8) | 0.14 |
| Nausea/vomiting | 79 (11.9) | 20 (13.6) | 59 (11.4) | 0.47 |
| Body mass index (IQR), kg/m$^2$ | | | | |
| Underweight (<18.5) | 22 (4.2) | 6 (5.0) | 16 (4.0) | 0.65 |
| Normal (18.5–22.9) | 223 (42.9) | 47 (38.8) | 176 (44.1) | |
| Overweight (23.0–24.9) | 157 (30.2) | 35 (28.9) | 122 (30.6) | |
| Obese I (25.0–29.9) | 94 (18.1) | 27 (22.3) | 67 (16.8) | |
| Obese II (≥30) | 24 (4.6) | 6 (5.0) | 18 (4.5) | |
| Median oxygen saturation (IQR), % | 97 (95–98) | 96 (93–97) | 97 (95–98) | <0.001 |
| Pneumonia on chest x-ray | 292 (67.1) | 86 (84.3) | 206 (61.9) | <0.001 |
| Disease severity at admission | | | | |
| Mild | 321 (48.3) | 42 (28.6) | 279 (54.0) | <0.001 |
| Moderate | 210 (31.6) | 56 (38.1) | 154 (29.8) | |
| Severe or critical | 133 (20.0) | 49 (33.3) | 84 (16.2) | |
| Outcomes | | | | |
| Died | 107 (16.1) | 40 (27.2) | 67 (13.0) | <0.001 |
| Alive | 557 (83.9) | 107 (72.8) | 450 (87.0) | |

*Patients may be in more than one category. Data are presented as numbers and percentages unless stated otherwise. Abbreviations: IQR = interquartile range.

Table 4 displays the association between DM and death in COVID-19 patients, controlling for variables existing before acquiring COVID-19. The univariate analysis showed that DM is significantly associated with mortality (OR 2.51; 95% CI 1.61–3.92). Moreover, other variables associated with death are age ≥60 years (OR 2.91; 95% CI 1.91–4.45), hypertension (OR 1.87;

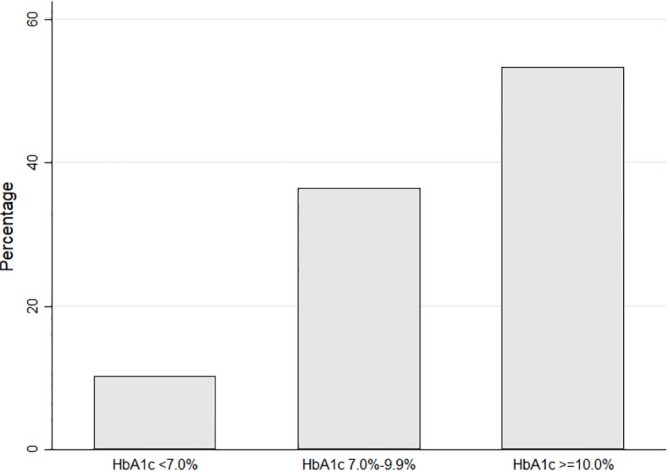

**Fig 2. HbA1c results of 118 COVID-19 patients with DM.**

95% CI 1.21–2.88), cardiovascular disease (OR 2.41; 95% CI 1.40–4.16), chronic kidney disease (OR 2.59; 95% CI 1.35–4.96), and neurologic disease (OR 4.79; 95% CI 2.08–11.02). After adjustment, age ≥60 years (aOR 2.21; 95% CI, 1.41–3.46) and DM (aOR 1.82; 95% CI 1.13–2.93) remained associated with mortality.

## Discussions

In an urban setting in Indonesia, we found that people with DM diagnosed with COVID-19 were admitted to the hospital with a high HbA1c value, which requires more attention. They are also more symptomatic and likely to present with comorbidities and severe disease. From the analysis, we found an association between DM and mortality. Another independent variable associated with death in our study is older age (≥60 years).

Literature has shown that poor glycaemic control in DM patients can lead to increased reactive oxygen species (ROS), pro-inflammatory cytokines, and disruption of various aspects of a person's immune response [5]. The increased expression of angiotensin-converting enzyme-2 (ACE2), a SARS-CoV-2 virus receptor, and a higher amount of intracellular furin in people with DM made it easier for the SARS-CoV-2 virus to enter cells and replicate, causing excessive inflammatory process, thereby increasing the morbidity and mortality of COVID-19 in

**Table 2. Laboratory parameters of the patients during the first week of hospitalization.**

| Laboratory results | Total (n = 664) median (IQR) | DM (n = 147) median (IQR) | Non-DM (517) median (IQR) | P-value |
|---|---|---|---|---|
| NLCR | 4.17 (2.33–7.09) | 5.61 (3.47–9.75) | 3.79 (2.18–6.50) | <0.001 |
| CRP, mg/dL | 3.45 (0.60–9.78) | 7.26 (3.45–12.74) | 2.25 (0.46–8.87) | <0.001 |
| D-dimer, ng/mL | 0.78 (0.41–1.74) | 1.02 (0.54–2.14) | 0.69 (0.38–1.55) | 0.002 |
| Ferritin, ng/dL | 560 (208–1256) | 817 (435–1431) | 520 (168–1225) | 0.002 |
| LDH, u/L | 289 (212–429) | 352 (256–444) | 283 (201–407) | 0.01 |
| RBG, mg/dL | 120 (103–159) | 238 (161–349) | 113 (99–132) | <0.001 |
| HbA1c, % | 9.3 (7.0–11.1) | 10.1 (8.2–11.3) | 6.4 (6.1–6.8) | <0.001 |

Missing data: NLCR (n = 29), CRP (n = 177), RBG (n = 107), LDH (n = 451), D-dimer (n = 214), Ferritin (n = 374). Abbreviations: IQR = interquartile range; DM = diabetes mellitus, NLCR = neutrophil-lymphocyte count ratio; CRP = C-reactive protein; LDH = lactate dehydrogenase; RBG = random blood glucose.

**Table 3. Patient's clinical characteristics difference between patients who died and who were discharged alive.**

| Characteristics | Died (n = 107) | Discharged alive (n = 557) | P |
|---|---|---|---|
| Male | 64 (59.8) | 293 (52.6) | 0.17 |
| Median age (IQR), years | 60 (53–70) | 49 (36–60) | <0.001 |
| Comorbidities | | | |
| No comorbidities | 28 (26.2) | 275 (49.4) | <0.001 |
| Diabetes mellitus | 40 (37.4) | 107 (19.2) | <0.001 |
| Hypertension | 42 (39.2) | 143 (25.7) | 0.004 |
| Cardiovascular disease | 22 (20.6) | 54 (9.7) | 0.001 |
| Chronic kidney disease | 15 (14.0) | 33 (5.9) | 0.003 |
| Liver disease | 2 (1.9) | 2 (0.4) | 0.06 |
| Malignancies | 2 (1.9) | 14 (2.5) | 0.69 |
| Chronic pulmonary disease | 6 (5.6) | 14 (2.5) | 0.09 |
| Neurological disease | 11 (10.3) | 13 (2.3) | <0.001 |
| Other comorbidities | 3 (2.8) | 22 (4.0) | 0.57 |
| Median symptom onset (IQR), days | 4 (2–10) | 14 (7–22) | <0.001 |
| Number of comorbidities in a patient | | | |
| No comorbidities | 28 (26.2) | 275 (49.4) | <0.001 |
| One comorbidity | 34 (31.8) | 189 (33.9) | |
| Two comorbidities | 33 (30.8) | 69 (12.4) | |
| Three comorbidities | 6 (5.6) | 19 (3.4) | |
| Four comorbidities | 5 (4.7) | 5 (0.9) | |
| Five comorbidities | 1 (0.9) | 0 (0.0) | |
| Body mass index (IQR), kg/m2 | | | |
| Underweight (<18.5) | 5 (6.5) | 17 (3.8) | 0.76 |
| Normal (18.5–22.9) | 35 (45.4) | 188 (42.4) | |
| Overweight (23.0–24.9) | 21 (27.3) | 136 (30.7) | |
| Obese I (25.0–29.9) | 12 (15.6) | 82 (18.5) | |
| Obese II (≥30.0) | 4 (5.2) | 20 (4.5) | |
| Median oxygen saturation (IQR), % | 94 (86–96) | 97 (95–98) | <0.001 |
| Pneumonia on chest x-ray | 62 (84.9) | 230 (63.5) | <0.001 |
| Disease severity at admission | | | |
| Mild | 17 (15.9) | 304 (54.6) | <0.001 |
| Moderate | 28 (26.2) | 182 (32.7) | |
| Severe or critical | 62 (57.9) | 71 (12.8) | |

Data are presented as numbers and percentages unless stated otherwise. Abbreviations: IQR = interquartile range.

people with type 2 DM [6–12]. Therefore, it has been suggested that DM can increase the risk of infection, hospital admission, severe disease, and death in patients with COVID-19. Nevertheless, the cause of more severe illness and higher mortality in COVID-19 patients with DM compared to the population without DM is multifactorial [13].

Hypertension, cardiovascular disease, and chronic kidney disease occur more often in people with DM in our study. This finding is similar to a study by Shi et al. and Cariou et al. in 2020 [14, 15]. In the Cariou et al. study, it was also found that of 1317 COVID-19 patients with DM, 47% already developed microvascular complications, and 41% had macrovascular complications [14].

COVID-19 patients presented with various symptoms in both groups. However, cough and dyspnoea were significantly more frequent in the DM group compared to the non-DM group.

Table 4. The association between diabetes and death.

| Variable | Category | Death n (row %) | OR (95% CI) | aOR (95% CI) |
|---|---|---|---|---|
| DM | No | 67 (13.0) | Ref | |
| | Yes | 40 (27.2) | 2.51 (1.61–3.92) | 1.82 (1.13–2.93)* |
| Sex | Female | 43 (14.0) | Ref | |
| | Male | 64 (17.9) | 1.34 (0.88–2.04) | 1.37 (0.89–2.12) |
| Age | <60 years | 53 (11.4) | Ref | |
| | ≥60 years | 54 (27.3) | 2.91 (1.91–4.45) | 2.21 (1.41–3.46)# |
| Hypertension | No | 65 (13.6) | Ref | |
| | Yes | 42 (22.7) | 1.87 (1.21–2.88) | 1.33 (0.83–2.13) |
| CVD | No | 85 (14.5) | Ref | |
| | Yes | 22 (29.0) | 2.41 (1.40–4.16) | 1.62 (0.89–2.95) |
| CKD | No | 92 (14.9) | Ref | |
| | Yes | 15 (31.2) | 2.59 (1.35–4.96) | 1.62 (0.81–3.23) |

*P = 0.01,
#P = 0.001.

A study in China during a similar period to ours has shown that dyspnoea (43%) and fatigue (50%) were more likely to be found in people with DM [16]. Whereas, a study of 154 COVID-19 patients with DM found that fever (81%) and cough (65%) were the most common symptoms [17].

Disease severity on admission also differs significantly between the two groups. Pneumonia on x-ray is more prevalent in people with DM, accompanied by lower median oxygen saturation. Thus, people with DM are more likely to present with more severe diseases. These findings agree with a study stating that people with DM were more likely to present with critical illness and pneumonia on the imaging on admission than those without DM [16].

In this study, all of the laboratory parameters of the patients are significantly higher in the DM group. Comparably, a study in Morocco in 2020 also found that people with DM had also experienced higher levels of several laboratory parameters. The study showed that CRP, ferritin, LDH, and D-dimer were significantly higher in the DM group [18]. Other studies further suggested that these parameters are associated with poor outcomes in COVID-19 patients [7, 19].

NLCR, as one of the crucial parameters in COVID-19 cases, is also significantly increased in the DM group. According to Shiny et al., an increase in NLCR strongly associates with glucose intolerance and insulin resistance in people with type 2 DM [20]. NLCR is one of the critical parameters in monitoring COVID-19 patients with DM because NLCR is reported in several studies to be associated with more severe clinical disease and worse prognosis [21–23].

In the univariable analysis, we found that DM, age, hypertension, cardiovascular disease, and chronic kidney disease are associated with death. In line with our study, several meta-analyses and a large study suggested that these variables are associated with COVID-19 mortality [24–26]. A study by Kim et al [24]. also showed that neurologic disorder is one of the factors associated with COVID-19 mortality in their multivariable analysis. We did not include neurologic disease in our multivariable model because we considered neurologic diseases in our setting (mainly stroke) to be in the causal pathway of mortality. However, in our multivariable analysis, we included the comorbidities related to DM and associated with death. After adjustment, the odds of death for COVID-19 patients with DM and older age remained significant. Comparably, several studies also showed that DM in COVID-19 patients was associated with

mortality [27, 28]. A retrospective cohort study in a tertiary hospital in India also generated a similar association between age and mortality in COVID-19 patients [29]. Furthermore, a large cohort study in Italy also suggested that increasing age was significantly associated with mortality in COVID-19 patients [30].

Compared to other studies investigating the association between DM and COVID-19 outcomes, this study provides more descriptive information about the laboratory characteristics of COVID-19 patients with DM in Indonesia, which could offer important knowledge for clinicians, especially for treatment considerations. Our study also only included patients with rRT-PCR results to minimize misclassification bias. Moreover, the sample size of our study is relatively large.

Our study has limitations; firstly, the data used in this study were retrieved during the first COVID-19 wave in Indonesia. Therefore, even though it gives valuable insight into how DM affects COVID-19 patients, vaccination and the spread of infection later may change the clinical characteristics of COVID-19 patients with DM. Secondly, we were unable to determine whether hyperglycaemia is a risk factor for COVID-19 or the other way around because we only have HbA1c data once they were diagnosed with COVID-19. Uncontrolled hyperglycaemia may be responsible for the change in the immune response, making people with DM more vulnerable to many infectious diseases, including COVID-19. However, infection is also known to be responsible for causing reactive hyperglycaemia, which may also explain the relatively high HbA1c levels among non-DM patients in this study, although these patients may also have undiagnosed DM or prediabetes.

## Conclusions

Our study found that COVID-19 patients with DM are likely to present with a very high HbA1c values, comorbidities and severe disease. Diabetic patients with COVID-19 are also more likely to experience worse laboratory results and death. The result of this study could guide clinicians to be more aggressive with the treatment when faced with COVID-19 patients with DM. Further studies with a larger sample size should be performed to determine the appropriate treatment modalities for COVID-19 with DM.

## Supporting information

**S1 File.**
(XLS)

## Acknowledgments

We thank R. Nina Susana Dewi, MD (Director of Dr Hasan Sadikin General Hospital), Laniyati Hamijoyo, MD, and all staff and residents contributing to the COVID-19 response team in Dr Hasan Sadikin General Hospital.

## Author Contributions

**Conceptualization:** Maya Kusumawati, Hikmat Permana, Nanny Natalia Mulyani Soetedjo, Begawan Bestari, Bachti Alisjahbana.

**Data curation:** Maya Kusumawati, Raspati Cundarani Koesoemadinata, Zuhaira Husna Fatma, Evan Susandi, Basti Andriyoko.

**Formal analysis:** Maya Kusumawati, Raspati Cundarani Koesoemadinata.

**Investigation:** Maya Kusumawati, Nanny Natalia Mulyani Soetedjo, Arto Yuwono Soeroto, Basti Andriyoko, Yovita Hartantri.

**Methodology:** Raspati Cundarani Koesoemadinata, Bachti Alisjahbana.

**Project administration:** Evan Susandi, Bachti Alisjahbana.

**Software:** Raspati Cundarani Koesoemadinata, Evan Susandi.

**Supervision:** Hikmat Permana, Nanny Natalia Mulyani Soetedjo, Arto Yuwono Soeroto, Begawan Bestari, Bachti Alisjahbana, Yovita Hartantri.

**Validation:** Basti Andriyoko.

**Writing – original draft:** Maya Kusumawati.

**Writing – review & editing:** Raspati Cundarani Koesoemadinata, Zuhaira Husna Fatma, Evan Susandi, Hikmat Permana, Nanny Natalia Mulyani Soetedjo, Arto Yuwono Soeroto, Begawan Bestari, Basti Andriyoko, Bachti Alisjahbana, Yovita Hartantri.

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
