## [Decision Letter · Decision Letter 0]

21 Mar 2023

PONE-D-22-31633The effect of diabetes mellitus on COVID-19 mortality among patients in a tertiary-level hospital in Bandung, IndonesiaPLOS ONE

Dear Dr. Koesoemadinata,

Thank you for submitting your manuscript to PLOS ONE. After careful consideration, we feel that it has merit but does not fully meet PLOS ONE’s publication criteria as it currently stands. Therefore, we invite you to submit a revised version of the manuscript that addresses the points raised during the review process.

Please describe how your study differs from others which have shown same association between Diabetes and COVID-19? State the strengths and limitations of your study in the discussion draft.==============================

We look forward to receiving your revised manuscript.

Kind regards,

Nosheen Nasir

Academic Editor

PLOS ONE

Journal Requirements:

Additional Editor Comments:

Please describe how your study differs from what is already reported about this association?

Reviewers' comments:

Reviewer's Responses to Questions

**Comments to the Author**

1. Is the manuscript technically sound, and do the data support the conclusions?

Reviewer #1: Yes

2. Has the statistical analysis been performed appropriately and rigorously? 

Reviewer #1: Yes

3. Have the authors made all data underlying the findings in their manuscript fully available?

Reviewer #1: Yes

4. Is the manuscript presented in an intelligible fashion and written in standard English?

Reviewer #1: Yes

5. Review Comments to the Author

Reviewer #1: The review was well written, clear, precise and to the point. The authors provided clear figures with legends, and overall the manuscript was easy to follow and understand. The discussion was elaborately done and overall, this was a good job.

6. PLOS authors have the option to publish the peer review history of their article (what does this mean?). If published, this will include your full peer review and any attached files.

Reviewer #1: No

---

## [Author Response · Author response to Decision Letter 0]

18 May 2023

Manuscript ID PONE-D-22-31633

The effect of diabetes mellitus on COVID-19 mortality among patients in a tertiary-level hospital in Bandung, Indonesia

Dear Prof. Nosheen Nasir and Reviewers,

Thank you for the helpful comments on our manuscript. We have now addressed the issue raised and edited the manuscript accordingly.

Editor comments:

Please describe how your study differs from what is already reported about this association. State the strengths and limitations of your study in the discussion draft.

We acknowledge that other studies have investigated the association between diabetes mellitus (DM) and COVID-19 outcomes. However, there are additional strengths of our research besides this association. We added, “Compared to other studies investigating the association between DM and COVID-19 outcomes, this study provides more descriptive information about the laboratory characteristics of COVID-19 patients with DM in Indonesia, which could offer important knowledge for clinicians, especially for treatment considerations. Our study also only included patients with rRT-PCR results to minimize misclassification bias. Moreover, the sample size of our study is relatively large.” (Lines 275-280). 

We also added, “Our study has limitations; firstly, the data used in this study were retrieved during the first COVID-19 wave in Indonesia. Therefore, even though it gives valuable insight into how DM affects COVID-19 patients, vaccination and the spread of infection later may change the clinical characteristics of COVID-19 patients with DM. Secondly, ...” (Lines 282-285).

1. Please ensure that your manuscript meets PLOS ONE’s style requirements, including those for file naming.

We have ensured that our manuscript meets the requirements.

2. In your Data Availability statement, you have not specified where the minimal data set underlying the results described in your manuscript can be found. PLOS defines a study’s minimal data set as the underlying data used to reach the conclusions drawn in the manuscript and any additional data required to replicate the reported study findings in their entirety. All PLOS journals require that the minimal data set be made fully available.

We have uploaded the minimal data set as a Supporting Information file.

The feedback is highly valuable, and we are grateful for your constructive input on our manuscript. Considering the raised concerns, we believe the paper has now improved.

On behalf of all authors,

Raspati Cundarani Koesoemadinata, MD, MSc

Research Center for Care and Control of Infectious Disease

Universitas Padjadjaran, Bandung, Indonesia

---

## [Editor Report · Decision Letter 1]

24 May 2023

The effect of diabetes mellitus on COVID-19 mortality among patients in a tertiary-level hospital in Bandung, Indonesia

PONE-D-22-31633R1

Dear Dr. Koesoemadinata,

We’re pleased to inform you that your manuscript has been judged scientifically suitable for publication and will be formally accepted for publication once it meets all outstanding technical requirements.

Kind regards,

Nosheen Nasir

Academic Editor

PLOS ONE
---

## [Editor Report · Acceptance letter]

6 Jun 2023

PONE-D-22-31633R1 

The effect of diabetes mellitus on COVID-19 mortality among patients in a tertiary-level hospital in Bandung, Indonesia 

Dear Dr. Koesoemadinata:

I'm pleased to inform you that your manuscript has been deemed suitable for publication in PLOS ONE. Congratulations! Your manuscript is now with our production department. 

Kind regards, 

on behalf of

Dr. Nosheen Nasir 

Academic Editor

PLOS ONE